# Conformation Variation and Tunable Protein Adsorption through Combination of Poly(acrylic acid) and Antifouling Poly(*N*-(2-hydroxyethyl) acrylamide) Diblock on a Particle Surface

**DOI:** 10.3390/polym12030566

**Published:** 2020-03-04

**Authors:** Zun Wang, Kaimin Chen, Chen Hua, Xuhong Guo

**Affiliations:** 1State Key Laboratory of Chemical Engineering, School of Chemical Engineering, East China University of Science and Technology, Shanghai 200237, China; wz18721289821@163.com (Z.W.); huachecust@163.com (C.H.); 2College of Chemistry and Chemical Engineering, Shanghai University of Engineering Science, Shanghai 201620, China

**Keywords:** diblock polymer brushes, pH stimuli response, turbidity titration, protein binding

## Abstract

Adsorption and desorption of proteins on biomaterial surfaces play a critical role in numerous biomedical applications. Spherical diblock polymer brushes (polystyrene with photoiniferter (PSV) as the core) with different block sequence, poly(acrylic acid)-b-poly(N-(2-hydroxyethyl) acrylamide) (PSV@PAA-b-PHEAA) and poly(N-(2-hydroxyethyl) acrylamide)-b-poly(acrylic acid) (PSV@PHEAA-b-PAA) were prepared via surface-initiated photoiniferter-mediated polymerization (SI-PIMP) and confirmed by a series of characterizations including TEM, Fourier transform infrared (FTIR) and elemental analysis. Both diblock polymer brushes show typical pH-dependent properties measured by dynamic light scattering (DLS) and Zeta potential. It is interesting to find out that conformation of PSV@PAA-b-PHEAA uniquely change with pH values, which is due to cooperation of electrostatic repulsion and steric hindrance. High-resolution turbidimetric titration was applied to explore the behavior of bovine serum albumin (BSA) binding to diblock polymer brushes, and the protein adsorption could be tuned by the existence of PHEAA as well as apparent PAA density. These studies laid a theoretical foundation for design of diblock polymer brushes and a possible application in biomedical fields.

## 1. Introduction 

Interaction between protein and functional biomaterials is of great importance due to a wide range of biomedical applications such as controlled drug release [1,2], gene delivery [3], biosensors [4], medical device coating [5] and so on. Especially, tunable protein adsorption by nanomaterials is essential in relevant research fields [6,7]. Polymer brushes acting as typical protein carriers have been widely studied in the past few years [8]. It is well-established that polymer brushes with core-shell structures are ideal models and promising candidates due to well-defined surface polymer chains and large surface-to-volume ratios [9], as well as superior stability [10]. Among them, diblock polymer brushes could combine two blocks with different structures and tune the comprehensive protein adsorption behavior as a whole [11,12,13,14]. Generally, protein adsorption to polymer brushes is a highly complicated process, which is often controlled by surface properties such as hydrophilicity/hydrophobicity; surface potential; roughness and external parameters, including pH, temperature, salt concentration, etc. [15,16,17,18,19]. It is not likely that a single factor is solely responsible for the adsorption behavior but, rather, a combination of many [20].

Poly(acrylic acid) (PAA) was generally used as the functional materials for protein adsorption. As a well-known pH-sensitive polyelectrolyte, PAA chains stretch out under high pH conditions due to electrostatic repulsion and shrink, with pH decreasing as a result of protonation [21,22,23]. It has also been used extensively to probe the interaction with proteins whose charges differ with the pH values. The interaction between them could be well-tuned by environmental parameters. Wang et al. prepared spherical PAA brushes to probe protein immobilization and separation and found that the amount of protein-binding to PAA chains would change with different pH values [24]. Meanwhile, Claus et al. reported that PAA brushes displayed a variable protein resistance that can be controlled by the ionic strength of the protein solution [15].

Compared to PAA with protein adsorption ability, some other polymers exhibit protein resistance in some extent. Hydrophilic poly(N-(2-hydroxyethyl) acrylamide) (PHEAA) is a typical antifouling material widely investigated [25,26,27]. The mechanisms for antifouling behavior of PHEAA have been proposed to interpret the complex process. One is hydration properties, and antifouling performance is positively correlated with the properties [11,28,29,30,31]. PHEAA achieve significant surface hydration via hydrogen bonds [32]. The other is a charge-charge interaction [33]. Generally, there is no electrostatic interaction between the protein and neutrally charged PHEAA. Therefore, PHEAA is regarded as one of the most potential antifouling materials [34]. 

However, little works have been focused on tuning protein adsorption through a combination of PAA and PHEAA in a polymer brush system. Here, we have synthesized diblock polymer brushes with these two polymers via surface-initiated photoiniferter-mediated polymerization (SI-PIMP). The photoiniferter is a sort of living radical initiator performing initiation, chain transfer and termination continuously, which allows the polymerization experiments to be conducted easily without toxic catalysts [35]. 

In this study, diblock polymer brushes with opposite block order poly(acrylic acid)-b-poly(N-(2-hydroxyethyl) acrylamide (PSV@PAA-b-PHEAA) and poly(N-(2-hydroxyethyl) acrylamide)-b-poly(acrylic acid (PSV@PHEAA-b-PAA) were obtained for the first time. Subsequently, TEM, Fourier transform infrared (FTIR), elemental analysis, dynamic light scattering (DLS) and Zeta potential were conducted to characterize the prepared materials. The conformation variation of diblock polymer brushes were investigated through DLS and element analysis, which could be elaborated by electrostatic repulsion and steric hindrance. Furthermore, the interaction between polymer brushes and BSA were studied by turbidimetric titration, respectively. This work provides new ideas to design new nanomaterials with controlled protein adsorption for biomedical applications.

## 2. Experiments

### 2.1. Materials

Styrene (St; 99%) and acrylic acid (AA; 99%) were purchased from Lingfeng Chemical Reagent Co., Ltd. (Shanghai, China) and purified by distillation under reduced pressure to remove the inhibitors prior to use. *N*-(2-hydroxyethyl) acrylamide (HEAA; 98%) was purchased from Tokyo Chemical Industry (Tokyo, Japan) and purified through alumina column chromatographic separation. Sodium dodecyl sulfate (SDS; 99%) was purchased from Amresco, Inc. (Boise, IA, USA). K_2_S_2_O_8_ (KPS; 99%) and BSA (98%) were purchased from J&K Chemical (Shanghai, China). Water used in this work was purified using reverse osmosis (Millipore Milli-RO, Darmstadt, Germany) and subsequent ion exchange (Millipore Milli-Q). All others were used as received.

Photoiniferter 4-vinylbenzyl N, N-diethyldithiocarbamate (VBDC) was obtained from the reaction of *p*-chloromethylstyrene (*p*-CMS; 99%; J&K Chemical, Shanghai, China) with sodium diethyldithiocarbamate (99%; J&K Chemical) according to the method reported previously [36]. Photoiniferter VBDC was measured by nuclear magnetic resonance spectroscopy (^1^H-NMR, Spectrometer AVANCE III 400, Bruker, Baltimore, MD, USA).

^1^H-NMR (CDCl_3_) results are: 7.35 (4H, d, Ar–H); 6.65–6.73 (H, q, =CH–Ar); 5.22–5.24 and 5.70–5.75 (2H, d, CH_2_=); 4.53 (2H, s, Ar–CH_2_–S); 4.03–4.15 and 3.71–3.73 (4H, q, –N(CH_2_CH_3_)_2_) and 1.26–1.30 (6H, t, –N(CH_2_CH_3_)_2_), which are same as the literature reported [37].

### 2.2. Synthesis of PS@VBDC Core Particles (PSV Core)

First, polystyrene (PS) seed particles were prepared by conventional emulsion polymerization within a 500-mL three-necked round-bottom flask equipped with a mechanical agitator and a nitrogen inlet [38]. St (10 g), SDS (0.2 g), KPS (0.578 g) and H_2_O (240 g) were added into the reactor and stirred at 300 rpm under nitrogen atmosphere for 1 h. Next, polymerization was carried out at 80 °C for 1 h. Finally, the PS seed particles were purified by dialyzing against pure water. PSV core particles were synthesized via seeded emulsion copolymerization. PS seed particles (15 g emulsion containing 0.5 g solid), KPS (0.05 g) and water (175 g) were placed in a 500-mL three-necked round-bottom flask and stirred at 300 rpm with a mechanical agitator for 30 min. Then, the flask was immersed into a 70 °C-oil bath under nitrogen protection. A mixture of St (0.8 g) and VBDC (0.25 g) was injected into the system (0.1 mL/min). The polymerization continued in the dark for 5 h. Finally, the PSV core particles were purified by dialyzing against pure water.

### 2.3. Synthesis of Diblock Polymer Brushes (PSV@PAA-b-PHEAA, PSV@PHEAA-b-PAA)

The diblock polymer brushes were prepared by surface-initiated photoiniferter-mediated polymerization. The AA monomer (0.4 g) and PSV core (75 g emulsion containing 0.38 g solid) were poured into a photoreactor equipped with a cold trap and a nitrogen inlet. The mixture was UV-irradiated at room temperature for 2.5 h with gentle magnetic stirring. The PAA-grafted PSV emulsion (PSV@PAA) was purified by dialyzing against pure water. Then, the reaction of the HEAA monomer (0.47 g) and PSV@PAA (42 g emulsion containing 0.24 g solid) was carried out under the conditions same as described in the above text. The obtained PSV@PAA-b-PHEAA were purified by dialyzing against pure water.

Similarly, HEAA monomer (0.75 g) and PSV emulsion (75 g) reacted to generate PHEAA-grafted PSV emulsion (PSV@PHEAA). PSV@PHEAA (40 g) and AA monomer (0.24 g) reacted to yield PSV@PHEAA-b-PAA. Basically, the mass ratio of PSV/AA is 1:1, PSV@PAA/HEAA is 1:2, PSV/HEAA is 1:2 and PSV@PHEAA/AA is 1:1.

### 2.4. Characterization

TEM, FTIR, elemental analysis, DLS and Zeta potential measurements were conducted as follows. The TEM images were performed by transmission electron microscopy JEOL-1400F (JEOL USA, Peabody, MA, USA). All samples were diluted with 1 mM PB buffer. A 20-μL solution was dropped onto a carbon film supported on a copper mesh grid and dried at ambient temperature overnight for TEM characterization. The contents of C, N and S elements were analyzed by ELEMENTAR vario ELIII (Elementar, Hanau, Germany). The experiment was carried out twice, and figures were averaged. All samples were lyophilized before analysis. FTIR spectra of the polymers were recorded on a Nicolet 6700 spectrophotometer, and all samples were frozen overnight and dried to a solid and mixed with KBr. The diameters and Zeta potential of particles were measured by PSS Nicomp 380 with the measurement angle of 173°. The polymer brush particles were diluted to 0.02 mg/mL in 1 mM PB buffer for size and Zeta potential measurements. Mean diameters were obtained from three duplicates.

Turbidimetric titration was also executed, and transmittance of polymer brush particles and BSA solution was monitored by a Brinkmann PC 950 colorimeter (420 nm filter) with a 2-cm path length optical probe while pH was recorded by an Orion pH electrode (ROSS Sure-Flow combination pH, 8172BNWP, Thermo Scientific Orion, Waltham, MA, USA). Both polymer brush particles and BSA were dissolved in 1 mM PBS buffer, and their concentrations were 0.004 and 0.02 mg/mL, respectively. The mixtures of BSA and polymer brushes were titrated from high to low pH. The turbidity was obtained from pH 10 to 2, which was achieved by monitorpolying the change of turbidity after continuous addition of 0.1 M HCl. Additionally, protein-free blanks were conducted in a similar way for a comparison.

## 3. Results and Discussion

### 3.1. Synthesis of Diblock Polymer Brushes

Two kinds of diblock polymer brushes were synthesized and fully characterized by TEM, FTIR, element analysis, DLS and so on. The morphologies of polymer brushes were obtained by TEM. It is obvious that all shapes of particles are spherical. A significant size distinction between PS seed and PSV core indicates photoiniferter VBDC and has been successfully grafted to PS seed. As shown in Figure 1, all kinds of polymer brush particles have almost similar sizes with the PSV core, because the brush structure with relatively low grafting density would collapse and hardly be observed in TEM images. On the other hand, DLS data shows diameters of PS seed and PSV core are 127 nm and 167 nm, respectively. Single-block polymer brushes PSV@PAA and PSV@PHEAA are 306 nm and 240 nm, and diblock polymer brushes PSV@PAA-b-PHEAA and PSV@PHEAA-b-PAA are 341 nm and 319 nm, respectively.

FTIR was applied to confirm the existence of functional groups (Figure 2). Compared to spectra of PS seeds, new absorption peaks for PSV at 1209 cm^−1^ and 1416 cm^−1^ are due to stretching vibrations of C=S and C–N in photoiniferter VBDC. The emerging peak at 1716 cm^−1^ was observed for PSV@PAA, PSV@PAA-b-PHEAA and PSV@PHEAA-b-PAA, which could be ascribed to the stretching vibration of C=O for carboxyl in PAA. Peaks at 1562 cm^−1^ (amide I) and 1651 cm^−1^ (amide II) for PSV@PHEAA, PSV@PAA-b-PHEAA and PSV@PHEAA-b-PAA result from stretching vibration of C=O and bending vibration of N–H, indicating the existence of HEAA structure.

Additionally, elemental analysis was accomplished to quantitatively examine the C, N and S components, as shown in Table 1. Firstly, the mass ratio of S/N in PSV core and PSV@PAA are 4.58 and 4.47, compared with the theoretical value of 4.57. Based on a four-layer model (Here, a, b, c and d are short for PS seed, photoiniferter VBDC, inner chains and outer chains), we could calculate the mass ratio of a:b:c:d in PSV@PAA-b-PHEAA and PSV@PHEAA-b-PAA, which are 6.59:1:1.48:3.37 and 6.59:1:2.22:2.32, respectively. Therefore, the molar ratio of AA/HEAA is 0.70/1.00 in PSV@PAA-b-PHEAA and 1.67/1.00 in PSV@PHEAA-b-PAA. Therefore, it is easily concluded that both diblock polymer brushes has the similar total contour length and almost the same block ratio (as illustrated in Figure A1).

### 3.2. Conformation Variation of Polymer Brushes

For the PSV@PAA-b-PHEAA system, the diameters of PSV@PAA range from 165 nm to 328 nm, while that of PSV@PAA-b-PHEAA vary from 227 nm to 356 nm as the pH increases, which both illustrate an obvious pH stimuli response owing to deprotonation of carboxyl groups in PAA chains. The pH range corresponding to the sharp size change is almost between 4 and 6 for PSV@PAA and PSV@PAA-b-PHEAA. Size of PSV@PAA is the same as that of a PSV core when pH is lower than 4, indicating complete collapse of PAA at this pH. It is worth noting that the size differences between PSV@PAA and PSV@PAA-b-PHEAA under various pH values are not a constant. In fact, the size differences turn larger at lower pH. It is likely that the PHEAA block is not fully stretched, especially when pH is larger than 4. From element analysis data, it is also noticed that the molar ratio of AA to HEAA in PSV@PAA-b-PHEAA is 0.7, which is remarkably different from the block length ratio of 5.9 at pH 10. It was noticed that when PAA chains shrink, the average distance between two neighboring PHEAA chains would decrease, resulting in stronger steric hindrance. Especially when pH drops from 5 to 4, PAA shrinks seriously, and a relatively stronger steric hindrance of PHEAA chains was produced, which causes their curly chains to stretch out. 

For the PSV@PHEAA-b-PAA system shown in Figure 3b, the size of PSV@PHEAA is around 245 nm with almost no response to pH change in the range of 3 to 10. Unlike the PAA block, no electrostatic repulsion exists in PHEAA, which is incapable of extending as pH changes. Qin et al. monitored the size change of PS@PHEAA with a wide range of pH, and no obvious size change was observed. Since PHEAA is a neutrally charged polymer, it does not interact with each other with electrostatic force, which indicates it has an excellent pH tolerance of 26. Furthermore, the molar content of PAA in PSV@PAA and PHEAA in PSV@PHEAA are very close, but the extended thickness of PAA (163 nm) is double that of PHEAA (80 nm) at pH 10, which demonstrates that electrostatic repulsion of PAA is much stronger than steric hindrance of PHEAA. PSV@PHEAA-b-PAA has a minimum diameter of 266 nm and a maximum diameter of 328 nm as pH changes from 4 to 10. Most of the size changes were observed at the pH range of 4–7. It was found that the diameter of PSV@PHEAA-b-PAA increases a little at pH 3. This is because electrostatic repulsion declines, leading to aggregation tendency at this pH, which is considerably distinct from PSV@PAA-b-PHEAA. Additionally, the apparent thickness ratio of AA to HEAA is 1.06, smaller than its molar ratio of 1.67, which could be explained by both weakening of electrostatic repulsion and steric hindrance in the outer layer.

Figure 3c depicts the PDI values under different pH for all kinds of polymer brushes. Average values for PSV@PAA, PSV@PAA-b-PHEAA, PSV@PHEAA and PSV@PHEAA-b-PAA are 0.023, 0.073, 0.073 and 0.056, which demonstrate great dispersity resulting from strict polymerization conditions. In addition, it turned out that PDI values do not reveal obvious dependence on the pH.

The surface charge density of the polymer brushes could be provided by Zeta potential, which has a direct relationship with the stabilization of particles in an aqueous environment [39]. Figure 3d presents the effect of pH on Zeta potential for all polymer brushes whose values are all negative. Firstly, Zeta potential of PSV@PAA-b-PHEAA is close to that of PSV@PAA. Zeta potential of both polymer brushes becomes stable when pH values are higher than 7. Most of the charge changes were observed in the pH range of 3–7, and Zeta potential declines with pH decreasing on account of the deprotonation of the PAA block. There is an intersection between Zeta potential of PSV@PAA and PSV@PAA-b-PHEAA. When pH is below 7, Zeta potential of PSV@PAA-b-PHEAA is smaller than PSV@PAA, which indicates that PHEAA chains would screen the anionic charge of PAA to some extent while Zeta potential becomes larger at pH higher than 7, which corresponds to the reduced apparent grafting density of PHEAA. Notably, PSV@PHEAA maintains a low charge in all pH values, though HEAA is electrically neutral, which could be ascribed to some residual SDS surfactant in the system. Secondly, Zeta potential of PSV@PHEAA-b-PAA is around −5 mV at pH 3, which explains the instability of the system, and changes to -22.66 mV at pH 7. Afterwards, there is no apparent change as the pH continues to increase, which is consistent with the diameter trend. It was calculated that the PAA content in PSV@PHEAA-b-PAA is 1.6 times of that in PSV@PAA-b-PHEAA. However, the Zeta potential of PSV@PHEAA-b-PAA is much smaller than PSV@PAA-b-PHEAA. It could be explained that the apparent surface charge density of PAA is smaller in PSV@PHEAA-b-PAA, because the PAA is in the outer layer. All polymer brushes with a PAA block indicate pH stimuli response properties.

### 3.3. Interaction Between Polymer Brushes and Protein 

To have a deep insight into the protein-binding behavior of diblock polymer brushes, the single block PSV@PAA and PSV@PHEAA were investigated as a start. Turbidimetry and DLS were combined to study the turbidity and size variation during the protein-binding process. BSA with an isoelectric point of 5 was chosen as a model protein in this work.

For the PSV@PAA and BSA system, the interaction process could be divided into 4 stages (pH 107, pH 75, pH 53 and pH 32) [19]. The turbidity remains near zero with a pH from 10 to 7, which indicates almost no interaction, because both protein and polymer brushes are negatively charged when Ph > pI, which is also proved by basically unchanged diameters, as illustrated in Figure 4a. After that, turbidity starts to increase at pH 7 where BSA begins binding to PAA chains. When the pH ranges from 7 to 5, positive patches on the BSA surface leads to adsorption regardless of its net negative charge [9]. The turbidity curve dramatically rises while diameters for the mixture have no increase, in contrast to the BSA-free system, which arises from that small-size BSA that penetrates into the polymer brush structure and contributes little in the size change. 

An interesting competition takes place from pH 5 to 3 in that protonation of PAA gives rise to a smaller size of PSV@PAA but BSA-binding makes diameters increase simultaneously. According to Figure 4a, the curve goes up. BSA adsorption is dominant at this stage. Notably, there is a 70 nm difference at pH 4, representing abundant BSA absorbs to the polymer brushes. We also found out the pH values corresponding to peaks of diameters and turbidity curves are not the same, which demonstrates that turbidity is much more sensitive to protein adsorption. When pH is lower than 3, the declines of both turbidity and diameter are attributed to the release of BSA from PSV@PAA. The PSV@PAA has a tendency to aggregate when pH is less than 3. 

Same measurements were performed for PSV@PHEAA. There is an average of an 11-nm diameter increase after BSA was added. The little increase may be caused by either negative charge of the PSV core or the entrapment of BSA by PHEAA chains [21]. Figure 4d shows that two curves of turbidity changes coincide, which proves that PHEAA almost does not interact with BSA.

After the second block PHEAA chains is grafted, a wider size change from pH 8 to 2 is exposed. The BSA-binding peak corresponding to a 60-nm-size increase appears at pH 4.6, as shown in Figure 5a, slightly larger than 4.3 in PSV@PAA curves. A significant discrepancy is that when pH is in the range of 3 to 2, both chains and PSV core adsorptions contributes to the increase of 33 nm size, which indicates PHEAA chains could impede desorption of BSA at low pH. Compared to the PSV@PAA system, the existence of PHEAA reduces the change of turbidity. Apparently, the area between turbidity curves of PSV@PAA-b-PHEAA in Figure 5b is smaller than that of the PSV@PAA system. In other words, PHEAA weakens the interaction with BSA to some extent at high pH and gets in the way of BSA release at low pH, which makes the process of BSA absorption and desorption become tenderer.

For the PSV@PHEAA-b-PAA system, there are some characteristics different from previous ones. Basically, no size change has been noticed in the pH range of 10 to 5, while the turbidity curve shows great change in pH 7 to 5. A peak was observed at pH 4, which has only a size difference of 28 nm from the blank system without BSA, considerably smaller than that of PSV@PAA and PSV@PAA-b-PHEAA, even though the length of PAA in the PSV@PHEAA-b-PAA system is longer according to calculations from the element analysis results. The reason is that the apparent grafting density is much smaller when PAA exists in the outer layer, which plays a pivotal role in BSA binding. In Figure 5d, turbidity curves also demonstrate the weaker interaction between PSV@PHEAA-b-PAA and BSA. As the pH gradually goes down, the size of the complex system keeps almost the same as that of the single-block polymer brushes, which is ascribed to no bound BSA under this condition.

## 4. Conclusions

In summary, two kinds of diblock polymer brushes (PSV@PAA-b-PHEAA and PSV@PHEAA-b-PAA) were synthesized by a surface-initiated photoiniferter-mediated polymerization (SI-PIMP). The molar ratio of the PAA/PHEAA block is 0.70/1.00 and 1.67/1.00 for PSV@PAA-b-PHEAA and PSV@PHEAA-b-PAA, respectively, based on the elemental analysis. The introduction of PHEAA in PSV@PAA-b-PHEAA could affect both the protein adsorption and protein release processes. PHEAA would hinder the penetration of the BSA into the PAA structure in some extent at high pH. On the other hand, PHEAA chains could stretch out and hinder the BSA release from the PAA structure at low pH. When PAA is grafted from the end of PHEAA in PSV@PHEAA, the apparent grafting density of PAA is reduced compared to that of PAA in PSV@PAA-b-PHEAA, and both the pH response and protein adsorption ability are reduced remarkably, while conformation of PHEAA does not change and contributes little to protein adsorption.

## Figures and Tables

**Figure 1 polymers-12-00566-f001:**
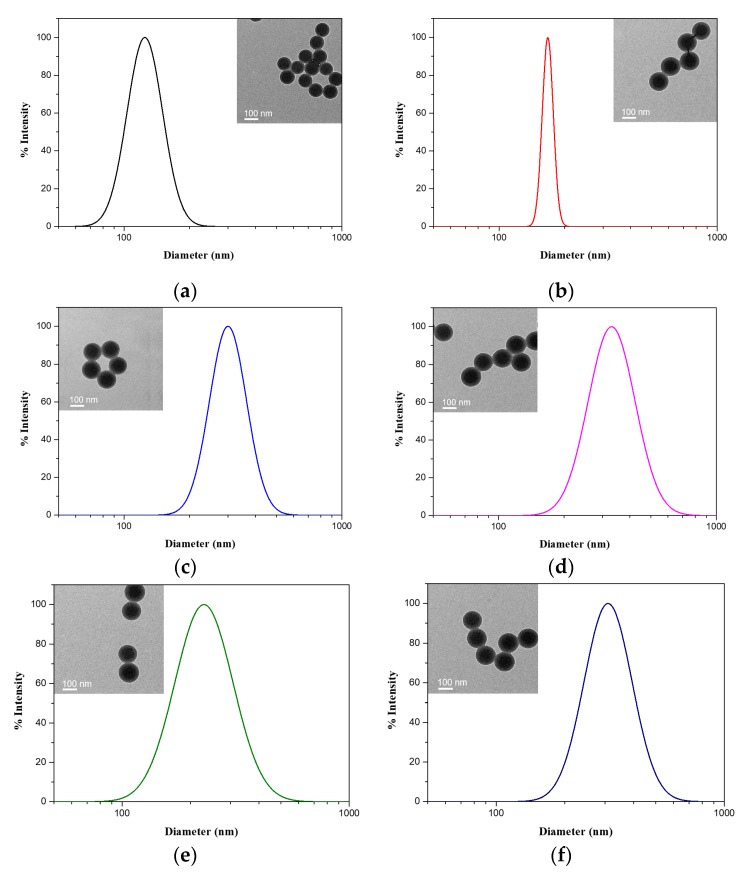
Size distributions and TEM images (insets) of polymer brushes. (**a**) polystyrene (PS) seed (black), (**b**) polystyrene with photoiniferter (PSV) (red), (**c**) PAA-grafted PSV emulsion (PSV@PAA) (blue), (**d**) poly(acrylic acid)-b-poly(N-(2-hydroxyethyl) acrylamide (PSV@PAA-b-PHEAA) (magenta), (**e**) PHEAA-grafted PSV emulsion (PSV@PHEAA) (olive) and (**f**) poly(N-(2-hydroxyethyl) acrylamide)-b-poly(acrylic acid (PSV@PHEAA-b-PAA) (navy).

**Figure 2 polymers-12-00566-f002:**
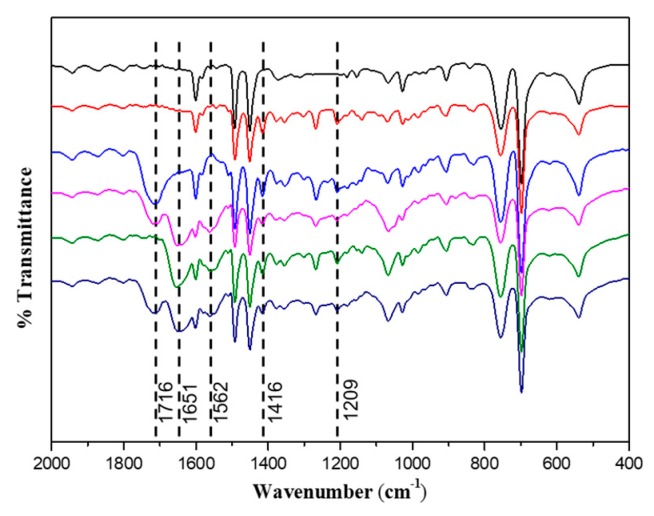
Fourier transform infrared (FTIR) spectra of polymer brushes. Line color denotation: PS seed (black), PSV (red), PSV@PAA (blue), PSV@PAA-b-PHEAA (magenta), PSV@PHEAA (olive) and PSV@PHEAA-b-PAA (navy).

**Figure 3 polymers-12-00566-f003:**
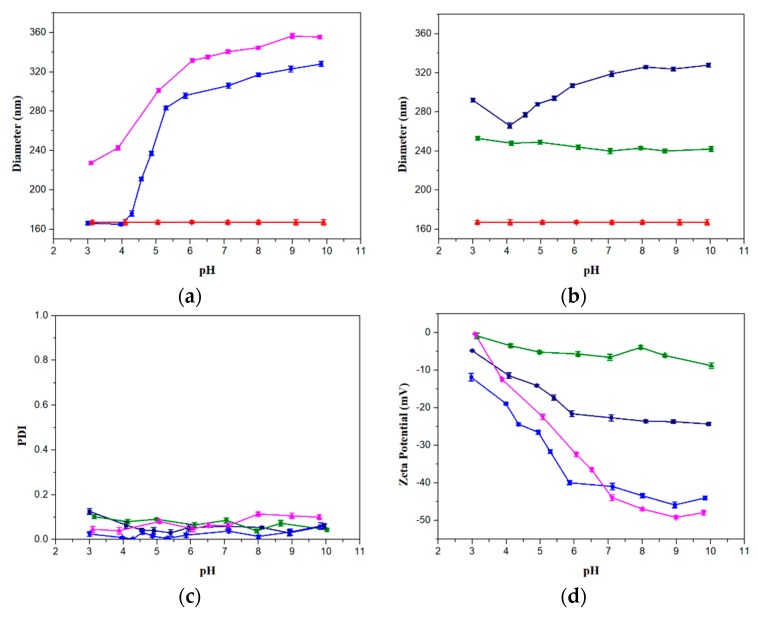
(**a**) Diameters for the PSV@PAA-b-PHEAA system, (**b**) diameters for the PSV@PHEAA-b-PAA system, (**c**) polydispersity index (PDI) and (**d**) Zeta potential of PSV@PAA-b-PHEAA and PSV@PHEAA-b-PAA systems from pH 3 to pH 10. Colors denote: red (PSV), blue (PSV@PAA), magenta (PSV@PAA-b-PHEAA), olive (PSV@PHEAA) and navy (PSV@PHEAA-b-PAA).

**Figure 4 polymers-12-00566-f004:**
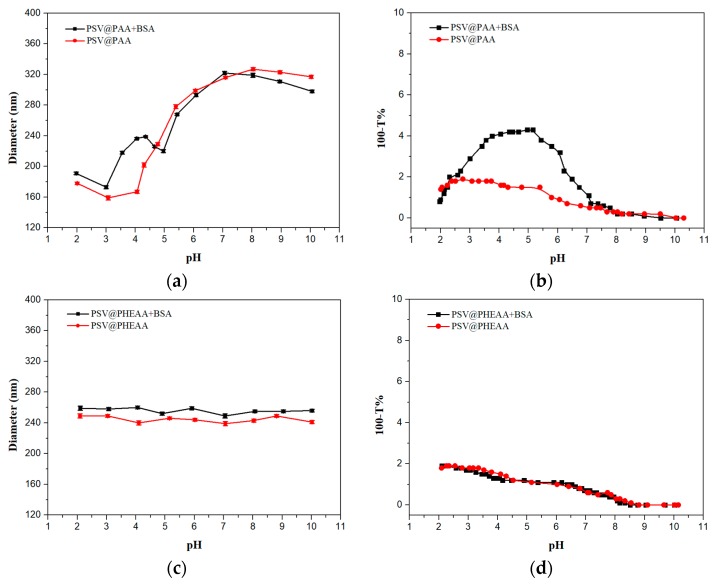
(**a**) Diameters and (**b**) turbidity of the mixture system of PSV@PAA with the BSA (
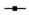
) and BSA-free system (
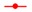
) and (**c**) diameters and (**d**) turbidity of the mixture system of PSV@PHEAA with the BSA (
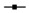
) and BSA-free system (
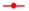
) from pH 2 to pH 10.

**Figure 5 polymers-12-00566-f005:**
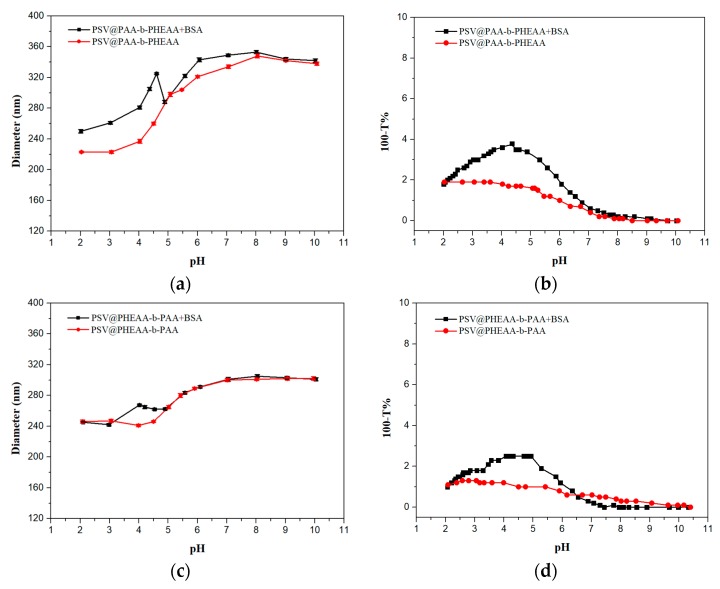
(**a**) Diameters and (**b**) turbidity of the mixture system of PSV@PAA-b-PHEAA with the BSA (
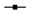
) and BSA-free system (
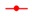
) and (**c**) diameters and (**d**) turbidity of the mixture system of PSV@PHEAA-b-PAA with the BSA (
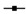
) and BSA-free system (
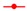
) from pH 2 to pH 10.

**Table 1 polymers-12-00566-t001:** Elemental analysis of diblock polymer brushes.

Samples	C%	N%	S%	S/N ^a^	S/N ^b^
PS	90.88				
PSV Core	87.40	0.70	3.18	4.58	4.57
PSV@PAA	84.59	0.60	2.66	4.47	4.57
PSV@PAA-b-PHEAA	78.48	2.08	1.94	0.93	–
PSV@PHEAA	80.98	2.07	2.46	1.19	–
PSV@PHEAA-b-PAA	79.12	1.85	1.99	1.08	–

^a^ Value from elemental analysis data. ^b^ Theoretical value. PS: polystyrene, PSV: polystyrene with photoiniferter, PSV@PAA: PAA-grafted PSV emulsion, PSV@PAA-b-PHEAA: poly(acrylic acid)-b-poly(N-(2-hydroxyethyl) acrylamide, PSV@PHEAA: PHEAA-grafted PSV emulsion and PSV@PHEAA-b-PAA: poly(N-(2-hydroxyethyl) acrylamide)-b-poly(acrylic acid).

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
