# Peer review of "Conformation Variation and Tunable Protein Adsorption through Combination of Poly(acrylic acid) and Antifouling Poly(N-(2-hydroxyethyl) acrylamide) Diblock on a Particle Surface"

_polymers, 2020, doi:10.3390/polym12030566_

Round 1
Reviewer 1 Report
The current manuscript by Wang et al. reports the effect of pH on conformation and protein adsorption behavior on block polymeric brushes of PAA and PHEMA. The study is interesting but requires major modifications before it could be possibly accepted for publication.
There are serious issues with the technical English along with the grammatical errors. I would advise that the authors get it corrected by some native English speaker. Line no. 53-55: “It has been recognized……adsorption”. It is still contradictory to say that hydrophobic surfaces exhibit protein resistance. Protein adsorption on solid surfaces is a highly complex process controlled by wettability, surface potential, roughness, pH, and salt concentration etc. and hence surface wettability alone doesn’t define rules for protein adsorption on surfaces. I would recommend that the authors reframe this section and include more relevant papers clearly stating that hydrophobicity alone cannot regulate protein adsorption. Few relevant references are listed below that can provide more insights into protein adsorption on surfaces.(a) J. Am. Chem. Soc. 1998, 120, 14, 3464-3473. (b) Advances in Colloid and Interface Sci. 2011, 162, 87-106. (c) Langmuir 2018, 34, 8178−8194. (d) ACS Biomater. Sci. Eng. 2018, 4, 9, 3224-3233. (e) J. Mater. Chem., 2007, 17, 4079-4087.
Line no. 56-58: “Furthermore, it is proved……………applications”. It is not clear what authors are trying to say. Amide groups are not necessarily responsible for the antifouling property of materials. In fact, it can enhance fouling due to ionic interactions between proteins and surfaces and also can attract bacteria (due to negative charge on the membrane) via opposite charges. Line no. 160-168: the entire paragraph should be reframed with more scientific soundness. Line no. 187-189: How do the variations in pH regulate the already adsorbed polymeric brush density? Please elaborate. I recommend that the authors include error bars for all the experimental data in all the reported graphs. It is very important to also include the significant difference to understand the process better. Line no. 195-200: Please elaborate and justify the behavior of PHEAA by citing relevant literature. Fig. 6a-b; Authors should clearly explain the differences observed between size and turbidity at pH 5. As per 6a, there is no interaction between BSA and particles then how come it shows maximum turbidity difference at the same pH. Proper justification is required.Author Response
Please see the attachment.

Reviewer 2 Report
Review comment to Polymer
Polymers encapsulate proteins are important for various applications. In this manuscript, author prepared two types of polymers based on the same monomers with different orientation and study their ability to carry BSA. The most interesting part is the encapsulation ability was reported to behavior different with different orientation, the
Although this is an interesting research topic, the logic of experimental design and the conclusion is not understandable for me. In this research, turbidimetric titration was used as major tool for the protein releasing. However, there are many other factors leading to the same observation. More argument or experiments are necessary for this concern. Moreover, more characterizations of the polymer products are necessary to support the conclusion.
Here are other suggestions
Why polymer-core shell structure are idea model for the purposes mentioned by authors? Moreover, I do not understand the relationship with the title of this research. Why the ration of composing elements conclude the ratio of two monomers? The figure 3 is not useful. The legends of figure need to improve. As example, “figure 3 (a)(b) diameter” what is this sentence for?Author Response
Please see the attachment.

Round 2
Reviewer 1 Report
The authors have defended the questions raised and successfully made the corrections as suggested. I would recommend acceptance of this manuscript in Polymers.
Author Response
Thanks very much for acceptance of this manuscript.
Reviewer 2 Report
Review comment to the polymer-696659 R1
I agree with most of the revision in this version of work except to the experiments for protein releasing and figure legends.
As mentioned in the previous comments, turbidity could not indicate the amount of protein. Many factors caused different turbidity and protein amount is only one of them. To conclude the protein releasing, suitable method for protein quantization might be necessary to support the result of turbidity. This must be clarify before publish
For the figure legends, more detail information was suggested. Meanwhile, figure 3 is not good and I still suggest redraw or delete.
Round 3
Reviewer 2 Report
Review comment to the polymers-696659
The only concern remains on the quantification of protein. After reading the reference author provided, there is no one using turbidity to estimate the amount of protein. I suggest author figuring this out.
Here are the method in those article
- Baier G.; Costa C.; Zeller A. BSA adsorption on differently charged polystyrene nanoparticles using isothermal titration calorimetry and the influence on cellular uptake. Macromolecular bioscience 2011, 11(5), 628-638.
I did not have access right to the first one.
- Inoue Y.; Ishihara K. Reduction of protein adsorption on well-characterized polymer brush layers with varying chemical structures. Colloids and Surfaces B: Biointerfaces 2010, 81(1), 350-357.
Measurement by QCM-D
- Sakata S.; Inoue Y.; Ishihara K. Molecular interaction forces generated during protein adsorption to well-defined polymer brush surfaces. Langmuir 2015, 31(10), 3108-3114.
The amount was estimated by SPR measurement
- Sigal G B.; Mrksich M.; Whitesides G M. Effect of surface wettability on the adsorption of proteins and detergents. Journal of the American Chemical Society 1998, 120(14), 3464-3473.
SPR measurement was used here
- Hasan A.; Pattanayek S K.; Pandey L M. Effect of Functional Groups of Self-Assembled Monolayers on Protein Adsorption and Initial Cell Adhesion. ACS Biomaterials Science & Engineering 2018, 4(9), 3224-3233.
Desorbed protein masses were quantified using QuantiPro BCA assay kit
Moreover, even those methods agreed possible error might generated from the degradation of protein.